# Review of Emerging Japanese Encephalitis Virus: New Aspects and Concepts about Entry into the Brain and Inter-Cellular Spreading

**DOI:** 10.3390/pathogens8030111

**Published:** 2019-07-26

**Authors:** Luis Filgueira, Nils Lannes

**Affiliations:** Platform Anatomy and Department of Oncology, Microbiology and Immunology, University of Fribourg, 1700 Fribourg, Switzerland

**Keywords:** Japanese encephalitis virus, flavivirus, JEV, epidemiology, symptoms, immune response, vaccine, blood–brain barrier, transmission, cellular events

## Abstract

Japanese encephalitis virus (JEV) is an emerging flavivirus of the Asia-Pacific region. More than two billion people live in endemic or epidemic areas and are at risk of infection. Recently, the first autochthonous human case was recorded in Africa, and infected birds have been found in Europe. JEV may spread even further to other continents. The first section of this review covers established and new information about the epidemiology of JEV. The subsequent sections focus on the impact of JEV on humans, including the natural course and immunity. Furthermore, new concepts are discussed about JEV’s entry into the brain. Finally, interactions of JEV and host cells are covered, as well as how JEV may spread in the body through latently infected immune cells and cell-to-cell transmission of virions or via other infectious material, including JEV genomic RNA.

## 1. Epidemiology of Emerging Japanese Encephalitis Virus

The Japanese encephalitis virus (JEV) is an emerging mosquito-borne flavivirus, related to dengue virus (DENV), West Nile virus (WNV), Zika virus (ZIKV), and tick-borne encephalitis virus (TBEV). JEV is an enveloped virus of about 50 nm virion size and consists of a positive-sense single-stranded RNA with a genome of about 11 kB with slight length differences between genotypes (Lu et al. 2017, Desingu et al. 2017) [1,2]. JEV is the major cause of encephalitis in humans in the Asia-Pacific region. Genetic studies propose that JEV evolved from an African ancestral virus that spread to the Indonesia-Malaysia region many centuries ago, from where it spread throughout Asia and over to Australia [3]. Serological and genetic research resulted in mapping one serotype and various genotypes throughout Australasia, as well as in Africa and Europe, also indicating continuous changes of geographic JEV distribution and JEV adaptation to new conditions [4,5,6,7,8,9].

To date, human JEV infections have been documented for most temperate, subtropical, and tropical regions of Asia, northern regions of Australia, and recently one case in Angola, Africa (Table 1 and Figure 1).

The tropical regions have been declared endemic, whereas regions with subtropical and temperate climate recorded epidemic events. Recently, a first autochthonous human case outside Australasia has been reported in Angola, Africa [10], and JEV has also been detected in birds in Italy, Europe [11]. However, various transmission chains and cycles have been described (Figure 2). Wild wading and water birds form a natural JEV reservoir, including migrating birds like herons, egrets, and ducks. Samy et al. (2018) modelled the distribution of egrets in the Australasian region, which overlaps well with the presence of JEV cases with the exception of the Middle East and parts of Australia, where the presence of the birds has been documented, but where no JEV cases recorded [7]. Thereby, one has to consider that herons and egrets migrate in summer for breeding to the northern temperate regions of Eurasia. They may also be all year round in the less cold and subtropical regions of Europe, Africa and Asia, but migrate to the tropical regions for wintering. Consequently, the long distance migrating birds (herons and egrets) may spread JEV across the sea and from endemic tropical areas into regions with subtropical and temperate climates where JEV may re-emerge. Ducks may spread JEV over shorter distances, but also across seas, as shown in Japan and Korea [12,13,14,15]. Interestingly, JEV has been found in the bone marrow of birds in Italy [11], probably due to long distance bird migration from Asia to Europe. However, little is known about migration routes of these birds. Technology, such as satellite-based tracking may provide more information through future studies to better estimate the spread of JEV by migrating birds, as it has been shown through recent studies for influenza H5N1 virus [16,17,18]. Most importantly, the infected birds are thought to be asymptomatic JEV carriers [9]. However, newer studies indicate that freshly hatched birds may become sick when infected with certain JEV strains [19]. More research is needed to better understand the immune biology of JEV in different susceptible birds. In addition to birds, bats may also be natural JEV reservoirs and may play an important role in spreading JEV [20]. However, very little is known about JEV infection in bats, as well as their migration habits. For the first African human case, the source of the infection and how JEV could have been transmitted are still not clear. Theoretically, there are two possibilities: firstly, migrating infected birds and carrier of JEV may have reached Angola, and then JEV may have been transmitted by corresponding local mosquitos. Secondly, one has to consider that Angola has become China’s major African trading partner, and China correspondingly Angola’s main trading partner, including for agricultural goods, also with corresponding exchange of workers from China and increased shipping of goods from China through tropical Southeast Asian waters to Angola. Asymptomatic JEV-infected carriers with high JEV titre, including wild rats or domestic animals, may have reached Angola, where local mosquitos may have then further transmitted JEV to the local human case. A similar, but inverted infection route from Angola to China has been recently described for yellow fever [21]. However, only new future cases and corresponding studies will clarify possible infection routes for African cases.

Feral and domestic pigs may play a peculiar role in the infection chain and cycle, and they are considered important carriers and amplifiers of JEV. Pigs are susceptible to JEV infection, but clinical symptoms in pigs are mild and when body temperature normalized, clinical symptoms decline and finally disappear [22]. However, piglets are stillborn when infected in utero, which indicates that JEV crosses the blood–placenta barrier in pigs [23]. Importantly, pigs become carriers of JEV and are able to transmit the virus among them in a vector-free mode through mucous contacts and corresponding micro-droplets [22,24]. Consequently, infected pigs may serve as a JEV reservoir, able to contribute to JEV circulation all year round in temperate climates and causing epidemic events in the warmer and wetter seasons when more mosquitos are around [25]. Increased pig farming with increased animal numbers worldwide may therefore increase the risk of spreading JEV and of causing epidemic infections in humans [23,24,26,27]. Recent experimental work in mice indicate that vector-free transmission of JEV may also apply to other species [28].

Culex and Aedes mosquitos are the JEV vectors, probably transmitting the infection from birds to susceptible other birds, mammals, and humans in the tropical and subtropical rainy season and in summer in areas with temperate climate [5]. Most important is the fact that the geographical distribution of potentially infected migrating birds, as well as of Culex and Aedes mosquitos, is much more extensive than the geographic area affected by JEV, including countries of the Middle East, Africa, the Americas, and Europe [4,5,29]. In addition, it is expected that the geographic areas with Culex and Aedes mosquitos will expand, and consequently new JEV epidemic regions will emerge, especially considering climate change with increased average temperatures in subtropical and temperate regions with increased humidity [26,30,31,32]. Spread of infected mosquitos in temperate regions may also increase by vertical transmission of JEV, as it would allow overwintering of the virus [26,33,34,35]. Interestingly, JEV carrying mosquitos have been found in Italy, Europe [36], indicating the potential of future autochthonous JEV cases in southern Europe.

Humans are dead-end hosts, as highly productive JEV viraemia is missing. Therefore, transmission from human to others via mosquitos is unlikely. However, cases of transmission through blood transfusions have been recorded [37]. In addition, JEV secretion by the mucous tissue of the throat has also been described in humans [38]. However, it remains an open question, whether mucous JEV production may be a source of vector-free JEV transmission between humans. It is also not known whether JEV infectious material is produced and secreted in the reproductive organs, as implicated for DENV [39], whether there was a risk of sexually transmitted disease in JEV infection, similar to ZIKV infection [40,41,42], or whether JEV crosses the placenta in humans, although it has been shown that JEV infects amniotic cells [43].

More than 2 billion people are living in endemic or epidemic areas and are thus at risk. The majority of JEV infected people develop mild symptoms that are common with many virus infections, or they may even be asymptomatic [44]. Therefore, it has been quite difficult to obtain exact numbers and assess the incidence of JEV infected people. In endemic regions, the annual infection rate of children has been calculated at about 5% [45]. About 30% to 70% of the non-vaccinated population in endemic areas are seropositive, indicating that they were infected at least once with JEV. Probably less than 1% of JEV infected humans develop disease. Especially children aged 0–14 years develop encephalitis, with about 50,000 to 175,000 cases per year, of which around 30% are fatal and 30 to 50% suffer from post-infectious neurological deficiencies. The majority of JEV infected people become immune, although possibly only against the one local genotype [45]. Recently, many cases have been recorded in vaccinated or immune populations, where JEV disease has resurged in the population above 15 years of age, probably due to introduction of new genotypes or due to the presence of non-immune individuals in the population [6,27,46].

## 2. Natural Course of JEV Infection, Virus Spread in the Human Body, Diagnostics, and Anti-Viral Treatment

Humans are usually infected through bites of JEV-infected mosquitos. Little is known about the early events after transmission, as incubation time and appearance of first symptoms take 5 to 15 days [34,47]. It has been proposed that the virus infects cells locally in the skin surrounding the mosquito bite, including fibroblasts [48], endothelial cells [43,49,50], pericytes [51], macrophages, and dermal dendritic cells [48,52,53,54], where a first round of virus amplification may take place. From there, the virus spreads to the brain, via newly produced virion particles, or by using migratory infected immune cells, including dendritic cells and T lymphocytes [55], which release infectious virions at their target location. Detailed information about virus entry into susceptible cells and early cellular response has been recently well reviewed by Yun and Lee (2018) [56] and is therefore not covered in this review. However, little is known about how JEV interferes with cellular structures and functions, which one would expected to happen through the non-structural (NS) proteins, microRNAs and fragments of its genomic and complementary RNA [57,58,59,60,61]. The mechanisms of how JEV gets into the brain and how JEV affects brain cells are explained in later sections. In contrast to DENV [62], little is known about how and whether JEV affects other body systems in humans, including the cardiovascular, digestive, reproductive, respiratory, and urinary system. However, mouse models indicate that JEV infects visceral organs in addition to the brain [63], whereby this viscerotropism possibly relates to the susceptibility of corresponding endothelial cells [49,64].

Laboratory diagnosis is essential to prove JEV infection and differentiate it from other flavivirus infections, as well as other causes of viral and bacterial encephalitis. Unfortunately, laboratory tests are usually done later in the course of the infection, when encephalitis has been diagnosed and the cause of it must be identified [65,66]. At that stage, an immune response is under way and anti-JEV IgM antibodies can be detected earlier, and IgG antibodies later in the course of the disease in the blood serum or the cerebrospinal fluid. For that purpose, various serological tests have been developed, including ELISA-based and haemagglutination inhibition (HAI) assays [38,67,68]. Unfortunately, specificity of serological tests is decreased due to cross reactivity with the different flaviviruses [66]. More recently, technically highly sensitive quantitative reverse transcription PCR assays have been established for the detection of genomic viral RNA in blood samples and cerebrospinal fluid, which differentiate between JEV genotypes and other flaviviruses [69,70,71,72]. Unfortunately, JEV genomic RNA is often not present in blood and cerebrospinal fluid, decreasing the overall sensitivity of the PCR assays.

When it comes to therapy, there is yet no drug or medication that can completely block JEV entry into cells or its replication in humans, or cure a JEV infection. However, Turtle and Solomon (2018) [47] have reviewed various possible therapeutic strategies. Furthermore, various compounds have been tested for anti-JEV antiviral effects, with some quite promising, including erythrosine B [73], tubacin [74], as well as aminoglycosides and tetracycline derivates [75]. Additional efforts have been put towards decreasing the inflammatory response and collateral damage in the JEV-infected brain to decrease neuronal cell death [76,77,78,79].

## 3. Immune Response and Vaccine

Humans usually respond with a strong physiological immune response to a first JEV infection, including the adaptive humoral and cellular immunity [80]. For a protective humoral immune response, generating neutralizing antibodies against the envelop (E) protein is essential, as the E protein is responsible for binding to and subsequent entry into the target cell [81,82]. Neutralising antibodies against JEV E protein may therefore be successfully used for passive immunisation in the early treatment of JEV-infected patients as indicated by animal experiments [82,83,84,85]. To date, 5 genotypes have been identified with variations of the E protein in the range of 1–5% [86,87], with corresponding variations in epitopes and resulting antibodies. However, infection or vaccination usually results in the generation of antibodies that cross-react with all JEV genotypes. Therefore, vaccines have been developed mainly against JEV genotype III with live attenuated or formalin-inactivated virus (e.g., IXARIO^®^), which induce long lasting immunity and are well tolerated by children and aged people [88,89,90,91,92,93,94,95]. Of note, JEV infection and vaccination may induce antibodies that are cross-reactive with other flaviviruses, including DENV [71,83,96,97]. Cross-reacting antibodies induced either through JEV infection or JEV vaccination may well have DENV infection-enhancement activity, resulting in fatal DENV infection outcome [98,99]. In pigs, antibody-dependent enhancement of JEV infection has also been shown after vaccination [100].

For building the specific humoral immunity against JEV, the adoptive T cell mediated immune response is required to provide support to the B cell development, including the process of somatic mutations for optimizing antigen binding properties and antibody class switch [101]. Consequently, a specific T cell response is essential in the course of JEV infection with possibly various effects by the T cell subpopulations [102,103]. Here, JEV-specific CD8+ cytotoxic αβ T cells, as well as γδ T cells, play a crucial role in limiting JEV production and spread by targeting and killing JEV-infected cells through cytotoxic mechanisms, thus preventing infection of the brain and subsequent encephalitis [104]. Those cytotoxic T cells recognise the NS proteins, of which immunogenic peptides—presented by MHC class I—have been identified [105,106]. JEV-specific CD4+ helper T cell recognising JEV E protein play a major role in providing B cell help in germinal centres. Otherwise, CD4+ cells recognising NS protein epitopes probably provide help to the specific cytotoxic immune response, produce anti-viral cytokines like interferon-γ, control and eventually resolve the cellular immune response, by also providing T cell memory [103,107,108,109].

Dendritic cells are essential for activation of T cells. They link the innate with the acquired immunity and they process and present JEV antigens on MHC class I and class II [110]. However, dendritic cells are also the target of JEV, resulting in productive infection [48,52,53,111]. Thereby, even the attenuated virus has been shown to infect dendritic cells [111]. In that respect, it remains unclear whether JEV infection of dendritic cells is important for building long-lasting immunity or whether infection of immune cells helps JEV to undermine an immune response or even leads to encephalitis.

In the first instance, mechanisms and cells of the innate immunity also contribute substantially to the recognition of the JEV infection and a protective immune response. Toll-like receptors (TLR), including TLR7 and TLR8, are important pattern recognition receptors sensing JEV components and alerting immune cells [77,112,113]. Natural killer cells (NK) and NK T cells (NKT) form the first cytotoxic line of defence until the acquired immunity takes over [52]. Mast cells, also belonging to the innate immunity, participate in the immune response against JEV. However, they may rather enhance JEV invasion and inflammation of the brain than contributing to its resolution [114].

Overall, JEV infection strongly activates all known immune processes and mechanisms, usually resulting in long-lasting immunity. However, more research is required for better understanding the interaction between JEV and the immune cells, especially in those cases that develop encephalitis.

## 4. JEV Crossing the Blood–Brain Barrier and Brain Infection

Crossing the blood-brain-barrier by JEV, infiltration and infection of the brain cells, as well as induction of inflammation resulting in encephalitis is still not well understood. Using mouse models, it has been shown that virus entry into the brain, infection of neurons and tissue-damaging inflammation anticipates the breakdown of the blood–brain barrier [115]. Therefore, we postulate two possible mechanisms of how JEV enters the brain tissue (Figure 3). First, endothelial cells of the brain capillaries may be infected with JEV, without being functionally affected, able to sustain the blood–brain barrier and subsequently pass the infection to underlying microglia cells and astrocytes, which pass it further on to neurons [77]. Our own research indicates that JEV infects and reproduces in endothelial cells but is not cytotoxic to endothelial cells of the microcirculation (manuscript in preparation). Thus, JEV virions circulating in the blood may directly infect the brain endothelial cells, or circulating infected immune cells, including monocytes, dendritic and T cells, may transfer JEV to the brain endothelial cells. Secondly, JEV-infected immune cells may enter through known physiological ways into the brain as in the healthy individual, that is, via the choroid plexus into the ventricular space from where they may spread the infection in the brain tissue [116]. Consequently, the breakdown of the blood–brain barrier may be only secondary to infection of nerve tissue cells and subsequent to the anti-viral and inflammatory response. In that respect, infected microglia cells and astrocytes may secrete damaging inflammatory factors, including interleukin-1β, interleukin-6 and tumour-necrosis factor α [77,117]. However, when it comes to the entry of JEV into the brain tissue, both ways discussed here could certainly apply.

Sometimes, JEV infection of the brain progresses in parallel with aseptic meningitis, which may affect the blood supply to the brain [118]. Usually, encephalitis is the most severe clinical appearance of JEV infection with a variety of first symptoms, including seizures, as well as acute sensory and neuromuscular functional deficiencies [119]. JEV infection of microglia [113,120], astrocytes [121] and neurons [122,123,124,125] with corresponding cellular response and possible apoptosis, as well as subsequent inflammatory response results in neuronal cell death. In addition, oedema and vascular damage may enhance damage to the tissue and cells. Accordingly, the basal ganglia, thalamus, and nuclei of the brainstem are most affected [126]. Japanese encephalitis is often focal affecting only one, or several brain regions and centres. If visual centres are affected, blindness may be the consequence. Deficiency of other sensory functions conducted through the cranial nerves and integrated in the centres of the midbrain and the brain stem may be targeted by JEV. Centres of vital functions in the brain stem may also be affected, including for respiration, cardiovascular and digestive regulation, resulting in corresponding acute symptoms like neurogenic respiratory deficiency, cardiovascular shock, and nausea. In addition, the motor neurons in the spinal cord may be affected, resulting in corresponding acute flaccid paralysis [118]. Substantial periventricular tissue damage may also result in an obstructive internal hydrocephalus with increased intracerebral pressure and subsequent additional brain damage [127].

Inflammation in the course of a JEV brain infection is due to an inflammatory reaction of the local microglia and astrocyte cell population, which may be infected or just respond to the surrounding cell and tissue damage in the course of the immune response against JEV-infected cells. These cells produce various inflammatory factors and cytokines with the aim to fight the infection. In addition, immune cells may be recruited from the periphery to the infected brain site, including inflammatory monocytes [76] and JEV-specific T cells [128], which will target infected local cells, possibly killing them and enhance inflammation, but also neuronal damage. However, in the best-case scenario, the immune response in the brain may clear the virus with minimal collateral damage. In the worst-case scenario, which applies to 20 to 40% of patients with encephalitis, neuronal JEV infection [123] and the immune response may damage key centres of the brain with long-term deficiencies or a fatal outcome [129,130].

Less severe JEV infection of the brain may result in transient milder symptoms, such as learning deficiency [131]. However, for 50% survivors of JEV brain infection, damage to the brain may result in a variety long-term syndromes, symptoms, and deficiencies, as compiled by Sarkari et al. (2012) [129].

Unfortunately, Japanese encephalitis can be reactivated in seropositive children, despite them having built efficient humoral anti-JEV immunity and without being newly infected. This may be due to chronic infection of T cells or microglia [113,132,133,134], indicating that those cells are long-term carriers and that those individuals have latent JEV infection. As the infectious JEV material hides in the cells, antibody-based immunity cannot help with controlling spread of JEV in the body, especially if the infectious material is transferred directly through cell-to-cell contact, with JEV-neutralising antibodies not being able to enter in contact with newly produced virions. Only efficient anti-viral drugs could interfere with intracellular JEV reproduction and interrupt re-activation of encephalitis and new lesions in the brain. Unfortunately, no such drug is yet available.

## 5. Cellular Events and Cell-to-Cell Transmission of JEV

As for all flaviviruses, JEV is persistent and non-pathogenic to susceptible cells of mosquitos. In that respect, one has to consider the long coevolution of flaviviruses in mosquitos with persistent infection [135]. Detailed information about virus-cell interaction in the mosquito has been covered by the review of Salas-Benito and De Nova-Ocampo (2015) [136].

Many different human cell types are susceptible to JEV, including immune cells (monocytes, macrophages, dendritic cells, microglia, T cells), astrocytes, neurons, endothelial cells, pericytes, and fibroblasts, but little is known about the interaction between JEV and human cells. However, each cell type reacts differently to JEV infection. In addition to methodological and technical experimental restrictions, the multifactorial variability of the host cells makes thorough understanding of JEV biology and pathology quite difficult. Furthermore, minor differences in the genome and consequently in the resulting proteins from the different JEV genotypes lead to slightly different cellular effects. For example, substitution of one amino acid in the M protein results in deficient assembling of infectious virions in mammalian cells, whereas the mutation does not affect production of infectious JEV in the mosquito [137].

Early cellular events, including viral entry and unfolding of viral control of the infected cell has been recently reviewed in general and with the focus on hepatitis C virus and DENV, by Neufeldt et al. (2018) [138], and for JEV by Yun and Lee (2018) [56]. Therefore, we cover here only mechanisms used by JEV to control infected human or mammalian cells and use them for its own purpose, as well as how JEV undermines some protective cellular responses. Thereby, information derived from other flaviviruses will also be considered.

Mammalian cells have evolved various mechanisms for the recognition of viral components, including TLR3, TLR7, and RIG-1 [139,140,141], as well as mechanisms (production of type I interferon) for subsequent blocking of viral survival and reproduction in the infected cell [142]. Toll-like receptor 7 (TLR7) plays thereby a crucial role for the recognition of intracellular JEV ssRNA in mammalian cells. TLR7 deficiency has been shown in a mouse model to result in decreased immune response with decreased type I interferon production and increased viral load in the brain [141]. Therefore, TLR7 is essential for building a protective immune response. However, TLR7 is up-regulated in the course of a JEV infection in human in immune cells, as well as in the brain, which may also result in an excessive and damaging inflammatory response and damaging encephalitis [141,143]. Unfortunately, the differential expression and regulation of TLR7 and other genes of early intracellular immune response during a JEV infection is not well understood, and it is not known whether and how JEV may interfere with this first step of induction of the immune response [77,142,144].

JEV uses and manipulates elements of the cytoskeleton, including actin, vimentin, and microtubules for cellular entry, transport of the various components for replication and assembly, as well as for release of infectious virions [145,146,147]. In addition, JEV also controls cell membranes and corresponding membrane proteins, that is, the cell surface membrane for viral entry, the endosomal membrane for viral fusion and delivery of the genomic RNA into the cytoplasm, the membrane complex of the endoplasmic reticulum for viral protein translation, and genomic RNA synthesis and finally assembly of the viral progeny [56,63,148,149,150]. However, the molecular mechanisms of how JEV interferes with cell membranes and the cytoskeleton are not well understood. Furthermore, JEV components enter the nucleus, where viral RNA synthesis has been described in insect and mammalian cells [151]. There, the virus may also influence transcription regulation of host cell genes that help to inactivate cellular anti-viral mechanisms. JEV may also prevent cell death and manipulate the cell cycle to keep the cell alive and in an optimal metabolic state, as long as viral reproduction is required [61,151,152,153,154,155,156,157,158,159]. JEV interferes with microRNAs to control the host cell [59]. In addition to the non-structural proteins, non-coding viral RNA of the positive and complementary genomic strain interfere with cellular processes [57,160,161]. However, more studies are required that investigate how and which host gene expression are manipulated by JEV.

In the process of JEV reproduction, synthesis of new genomic RNA is essential. Subsequently, this RNA is packed into virions that are able to infect other cells, when released into the extracellular space. Interestingly, plain genomic JEV RNA is sufficient for infection of a cell [162]. Our most recent research indicates that transmission of JEV between cells may not need assembly of virions, but single stranded or double stranded genomic JEV RNA may be transferred from one cell to another [163]. The mechanisms for that are still under investigation. However, exchange of infectious material between cells may spread JEV in certain tissues (e.g., the brain). One possible mechanism for spreading could be the transfer of infectious material through gap junctions, which are plentiful in the nervous tissue connecting neurons, but also astrocytes and microglia, as well as connecting all three cell types with each other [164,165]. Exchange of JEV genomic RNA through gap junctions may just work for the diameter of about or smaller than 2 nm [166]. It has already been shown for exchange of RNA in cardiomyocytes [167] and it has been postulated for osteocytes in bone [168]. Such JEV spreading mechanism would explain the focal appearance of JEV-induced brain lesions and re-activation of encephalitis in seropositive children with latent JEV infection [132]. Similar findings have already been published, showing that infectious material could also be transferred between cells through tunnelling effects or nanotubes [169,170,171] or via extracellular vesicles [172]. However, more research is needed to investigate the various possible ways of exchange of infectious JEV material between cells. Better understanding of those mechanisms will then allow a targeted development of new drugs preventing JEV spreading in the body, tissue and between cells in a newly infected individual or in patients with latent infection.

## 6. Summary and Conclusions

The first part of this review summarized essential epidemiologic information about the emerging JEV emphasizing on the increasing worldwide threat that JEV poses to humans, especially considering global warming and the corresponding geographic expansion of mosquitos that are JEV vectors. In addition, the role of increased world trade with corresponding international exchange of workers and animals was briefly mentioned, as well as bird migration routes of corresponding JEV infected birds as carriers were discussed as unmanageable risk of geographical spreading of JEV. The subsequent sections discussed biological mechanisms that contribute to JEV infection and pathology focusing on human. Thereby, spread of virus in the human body were described and new concepts of JEV entry into the brain were proposed. In addition, the interaction between JEV and various cell types and the immune system were elucidated, also connecting the clinical appearance of JEV infection with basic scientific concepts. The most recently discovered way of JEV transmission between cells was also discussed as a new concept. Thereby, intercellular JEV transmission may happen via plain genomic JEV RNA, without forming complete virions. This new concept may well explain re-emerging of JEV in infected and immune individuals, in the absence of reinfection. It may also explain the focal spreading of JEV infection in certain brain areas. However, a better understanding of JEV infection at cellular and system level is required for the targeted development of new efficient anti-viral treatments.

## Figures and Tables

**Figure 1 pathogens-08-00111-f001:**
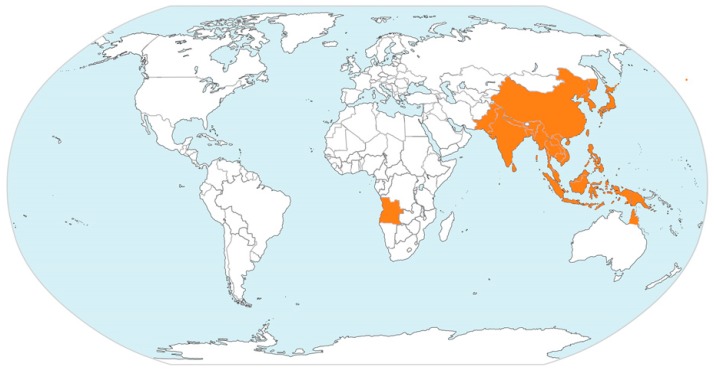
Geographic regions and countries with documented human JEV cases, modified from https://www.cdc.gov/japaneseencephalitis/Maps/index.html.

**Figure 2 pathogens-08-00111-f002:**
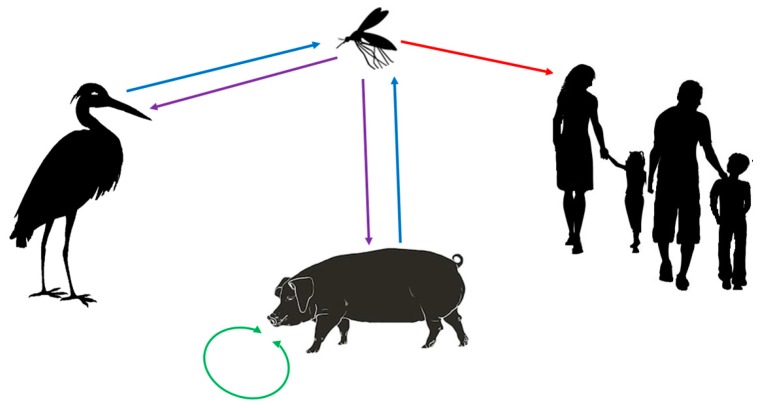
Infection chain and cycles of JEV. Wading and water birds are natural hosts and carriers of JEV. Culex and Aedes mosquitos transfer JEV between birds and to other species. Through seasonal migration, the birds spread JEV between tropical, subtropical, and temperate regions and across the sea. Wild and domesticated pigs play a special role in the infection chain, as they are carriers and amplifiers of JEV through vector-free transmission of JEV between them. Humans are dead-end hosts with about 1% succumbing to encephalitis, usually with subsequent severe deficiencies or fatal outcome.

**Figure 3 pathogens-08-00111-f003:**
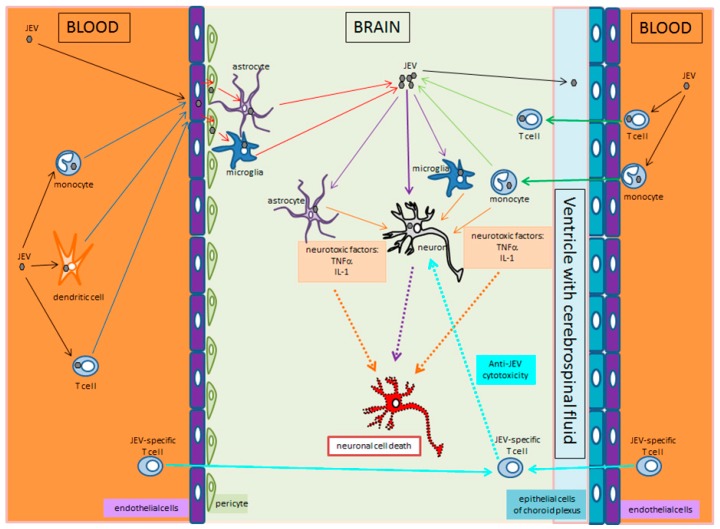
JEV enters the brain through two different ways and leads to infection of neurons and encephalitis. On the left, it is proposed that JEV or JEV-infected leucocytes (monocytes, dendritic cells, T cells) circulating in the blood infect endothelial cells of the brain capillaries. The endothelial cells, without being harmed by JEV, amplify and transmit JEV to pericytes, or even microglia cells or astrocytes that are in contact with the capillaries. Infected pericytes, microglia, and astrocytes amplify JEV and transmit it to other brain cells, including neurons, astrocytes, and microglia. On the right, JEV-infected T cells or monocytes circulating in the blood migrate through the choroid plexus into the ventricular space with its cerebrospinal fluid, and from there into the periventricular nervous tissue. There, microglia, astrocytes, and neurons are infected either by cell-to-cell contact, or by newly produced extracellular JEV virions. In the middle, JEV-infected neurons undergo eventually apoptosis. In addition, JEV-infected microglia and astrocytes produce inflammatory factors (TNFα, IL-1) that induce collateral damage resulting in apoptosis of non-infected neurons.

**Table 1 pathogens-08-00111-t001:** Countries and Geographic Regions with documented JEV cases.

Geographic Region	Countries/Territories/Provinces
**East Asia**	China (including Tibet)
	Japan
	North Korea
	Russia (Far East Provinces)
	South Korea
	Taiwan
**South Asia**	Bangladesh
	India
	Nepal
	Pakistan
	Sri Lanka
**Southeast Asia**	Brunei
	Cambodia
	Indonesia
	Laos
	Malaysia
	Myanmar
	Papua New Guinea
	Philippines
	Singapore
	Thailand
	Timor-Leste
	Vietnam
**Australia**	Cape York Peninsula of Queensland
	Top End of Northern Territory

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
