# Peer review of "Review of Emerging Japanese Encephalitis Virus: New Aspects and Concepts about Entry into the Brain and Inter-Cellular Spreading"

_pathogens, 2019, doi:10.3390/pathogens8030111_

Round 1

Reviewer 1 Report

The manuscript "Review of emerging Japanese encephalitis virus: established and new aspects and concepts" provides an overview of selected topics of that virus. The review seems a little unfocused from the title, which does not clearly reflect the subject of the review, the abstract that again does not  highlight the contents and the main body of the manuscript, which does not provide a thorough consideration of the topics selected. This detracts from what could be a useful addition to the subject. The following should be considered by the authors:

Decide what aspect of JEV the review will focus on, epidemiology, pathogenesis, diagnosis, entry into the CNS (this appears to be the authors area of experience). Then provide a well rounded discussion of that topic rather than just reporting what has been published without some evaluation. For example, the authors state three times within the first two pages that JEV has been detected in Africa and Europe without any consideration of the strength or accuracy of these reports. Why have there been no further cases?

Revise the title. Give an indication for the reader of the subject. 

The abstract should be a structured paragraph summarising the main points of the manuscript. What are the 'various aspects' and 'new biological concepts' that will be considered.

Lines 34-39 are a list of countries, not a paragraph, suggest a table.

The review desperately needs a Conclusion section that highlights the novelty of the review and/or suggests future research.

Minor comments - the choice of phrases could be improved:

Why use small to describe a virus, relative to what?

Why not be precise about the length of the virus genome?

Lines 48/49, rephrase the sentence to be more precise about how migrating birds could spread JEV beyond current endemic areas.

Line 64, how do the authors define something as 'most important'?

Line 73, JEV viraemia in humans is not missing, it is lower than that needed to infect a mosquito feeding on the human.

Line 78, "as implicated" not as 'shown'.

Line 82, are 2 billion people 'leaving' or "living"?

Line 84, clarify this statement, it is easy to calculate but the accuracy is debatable.

Line 110, use "in later sections" not 'further down'.

Line 116, "as other causes of virus and bacterial encephalitis."

Lines 123 - 127, the sentences need to be rephrased "detection of genomic viral RNA", "which differentiate between", "RNA is often present".

Line 130, "therapeutic strategies".

Line 127, "Humans usually respond"

Line 153, assume you mean "adaptive" not adoptive.

Line 188, "without being functionally"

Line 191, "Our own" 

Line 196, what are 'physiological ways'?

Line 234, what does "or just respond to the surround cell and tissue damage." mean?

Line 243, "such as" not 'like'.

Line 245, "syndromes"

Line 263, what does 'somehow controls', this imprecise phrase is typical and hardly represents a New Aspect of JEV infection.

Line 269, what is 'counter fights'?

Line 278, suggest "Our most recent research" rather than 'newest'.

Line 284, what is "Exchange of thin JEV genomic RNA"?

Author Response

We thank reviewer 1 for assessing the manuscript and we are grateful for the valuable feedback.

We have changed the title and the abstract according to the request and hope that they are now more focused on the main message of the manuscript.

In addition, we have corrected all suggested English writing mistakes and also asked an native English-speaking academic to give feedback on English writing.

All changes have been marked in yellow.

Reviewer 2 Report

In this review, Filgueira and Lannes described the epidemiology, sypmtoms caused by JEV, immune response, and development of vaccine, transmission and the cellular events of JEV.

Overall, this review is well written and covers many aspects of JEV, except for the cellular events section. I understand that the author wants to discuss the intracellular response of Japanese encephalitis virus infected with different cell lines, but each cell is not responded to by the Japanese encephalitis virus infection. Make each of the discussion results of this article not a concrete conclusion. Furthermore, I suggest that authors should point out the cellular events of JEV-oriented is in the human cells or mosquito cells. For instance, JEV may prevent cell death and manipulate the cell cycle to keep the cell alive…….this result is only demonstrated in the mosquito cells but not human cells. Please specify which cell type when you discuss the cellular events.

Author Response

We thank reviewer 2 for assessing our manuscript and for his valuable feedback.

We have rewritten most of the last section to better clarify the difference between the interaction between JEV and insect cells, or JEV and mammalian cells, respectively.

Thereby, one has to consider that in the references that we have mentioned, JEV has been shown to interfere with cell survival in both, insect and mammalian cells, respectively (Weng JR, Hua CH et al., 2018; Suzuki T, Okamot T et al., 2018; Vincenzi E, Pagani I et al., 2018).

The changes and additions have been marked in yellow.

Round 2

Reviewer 1 Report

The revised review "Review of emerging Japanese encephalitis virus: new aspects and concepts about entry into the brain and inter-cellular spreading" is a considerable improvement. The authors would be advised to give it a final edit to improve the language throughout, for example line 75 'climate record epidemic events'. Line 90 'Technology, such as satellite-based tracking...'. Line 95 '..[19]. More research is needed...'. and so on.

On that point, the authors over use the word interestingly. That is usual the preserve of the reader to define what is and is not interesting. Secondly, the authors overuse the phrase 'more studies/research are needed'. Once, maybe twice but no more.

The review needs a short conclusion section, this should be included to bring the review to a close rather than the sudden finish on line 399.

Author Response

We thank the reviewer for his/her valuable feedback. All of it has been considered. Spelling, punctuation and style has been rechecked, especially the problems mentioned by the reviewer. A last section has now been included with the subtitle ¨Summary and conclusion¨.